# From Novelty to Imitation: Self-Distilled Rewards for Offline Reinforcement Learning

**Gaurav Chaudhary**  *gauravch@iitk.ac.in*
*Department of Electrical Engineering*
*Indian Institute of Technology Kanpur*

**Laxmidhar Behera**  *lbehera@iitk.ac.in*
*Department of Electrical Engineering*
*Indian Institute of Technology Kanpur*

**Reviewed on OpenReview:** *https://openreview.net/forum?id=F5K94JI2Jb*

## Abstract

Offline Reinforcement Learning (RL) aims to learn effective policies from a static dataset without requiring further agent environment interactions. However, its practical adoption is often hindered by the need for explicit reward annotations, which can be costly to engineer or difficult to obtain retrospectively. To address this, we propose ReLOAD (**Re**inforcement **L**earning with **O**ffline Reward **A**nnotation via **D**istillation), a novel reward annotation framework for offline RL. Unlike existing methods that depend on complex alignment procedures, our approach adapts Random Network Distillation (RND) to generate intrinsic rewards from expert demonstrations using a simple yet effective embedding discrepancy measure. First, we train a predictor network to mimic a fixed target network's embeddings based on expert state transitions. Later, the prediction error between these networks serves as a reward signal for each transition in the static dataset. This mechanism provides a structured reward signal without requiring handcrafted reward annotations. We provide a formal theoretical construct that provides insights into how RND prediction errors effectively serve as intrinsic rewards by distinguishing expert-like transitions. Experiments on the D4RL benchmark demonstrate that ReLOAD enables robust offline policy learning and achieves performance competitive with traditional reward-annotated methods.

## 1 Introduction

Deep Reinforcement Learning (DRL) has achieved remarkable success across a wide range of applications, from game-playing (Silver et al., 2017; Vinyals et al., 2019) and intelligent perception systems (Chaudhary et al., 2023) to financial modeling (Charpentier et al., 2021). Despite these breakthroughs, DRL methods typically require millions of state-action transitions to train effective policies. This sample inefficiency makes DRL impractical for many real-world scenarios and motivates the search for alternative learning paradigms.

Offline reinforcement learning (RL) is a powerful approach to mitigate these challenges by leveraging static datasets to train policies without real-time interactions. This is particularly advantageous in areas such as robotics (Tang et al., 2024), autonomous driving (Kiran et al., 2021), and healthcare (Yu et al., 2021), where collecting data through trial and error may be prohibitively expensive, unsafe, or simply unfeasible. However, a common requirement across many offline RL methods is the need for explicit reward annotations. In many practical settings, these reward signals are either scarce or costly to obtain. While techniques like Aligned Imitation Learning via Optimal Transport (AILOT) (Bobrin et al., 2024) have attempted to overcome this limitation by aligning trajectories in latent intent spaces, such methods tend to incur significant computational overhead and can struggle when faced with the multi-modality of intents.

To address these concerns, we present ReLOAD (**Re**inforcement **L**earning with **O**ffline Reward **A**nnotation via **D**istillation), a novel framework that leverages Random Network Distillation (RND) (Burda et al., 2018) to generate intrinsic rewards using the discrepancy between the embeddings of the target and predictor networks. Unlike prior methods that require adversarial training, trajectory alignment, or latent space modeling, ReLOAD only requires standard forward passes through neural networks, making it simple to implement and highly scalable. Furthermore, our use of RND departs from prior novelty-based applications in offline RL (Nikulin et al., 2023) by re-framing it as a reward inference mechanism grounded in expert imitation. Rather than discouraging novelty, ReLOAD distills prediction error from expert-only state-transition pairs (s, s') to promote expert-aligned behavior, in contrast to prior work, which operates on state-action pairs (s, a) to penalize out-of-distribution actions. This conceptual shift—formalized through our theoretical analysis—enables scalable and effective reward inference under minimal supervision.

ReLOAD circumvents the reliance on manually engineered reward signals by training a predictor network to mimic the embeddings produced by a fixed, randomly initialized target network using expert transitions. After training the predictor to match the embedding of the target network over expert transitions, the prediction error between the predictor and target networks for each state transition in the offline dataset serves as an intrinsic reward. In this formulation, small errors indicate behavior similar to expert transitions, while larger errors highlight novel or suboptimal transitions. This simple yet effective mechanism facilitates robust and automated reward generation without incurring the heavy computational costs of complex alignment procedures.

Our method contributes to the field in several key ways:

- **A Novel Reward Distillation Strategy:** We introduce a technique that bypasses the need for handcrafted rewards by inferring intrinsic rewards from RND-based embedding discrepancies, leveraging accessible expert demonstrations.

- **Theoretical Foundation for RND Rewards:** We rigorously establish a formal theorem demonstrating that RND prediction errors are a reliable intrinsic reward signal, effectively distinguishing expert-like transitions in offline RL.

- **Implicit Transition Quality Assessment:** ReLOAD is able to assess the quality of transitions without requiring explicit action-specific labels, thereby differentiating expert behavior from suboptimal strategies.

- **Competitive Benchmark Performance:** Extensive evaluations on the D4RL benchmark (Fu et al., 2020) show that ReLOAD performs competitively with traditional reward-based offline RL approaches.

Compared to existing reward inference techniques, such as Optimal Transport Reward (OTR) labeling (Luo et al., 2023), AILOT (Bobrin et al., 2024), and Calibrated Latent Guidance (CLUE) (Liu et al., 2023), ReLOAD distinguishes itself through its conceptual simplicity and computational efficiency. Prior approaches (OTR and AILOT) depend on expensive trajectory alignment or latent space optimization methods that may falter with multimodal expert data or depend on expert action labels (CLUE). However, our framework provides intrinsic reward annotation using straightforward embedding discrepancies between state transitions. This makes ReLOAD particularly well-suited for large-scale and diverse offline datasets where both computational resources and adaptability are critical.

Drawing on insights from offline imitation learning (Kostrikov et al., 2021), optimal transport-based reward inference (Luo et al., 2023), and novelty-driven exploration (Yang et al., 2024), ReLOAD offers a practical and scalable solution to the reward scarcity problem inherent in offline RL. In the following sections, we detail our methodology, present comprehensive experimental evaluations, and discuss the broader implications of our findings for advancing the field of offline reinforcement learning.

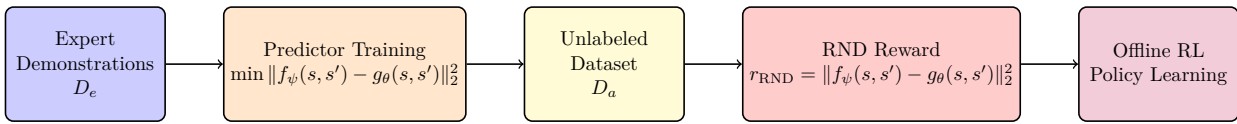

Figure 1: Overview of our method, ReLOAD. Transitions from expert demonstrations are used to train a predictor network to align with a fixed target network. When the unlabeled transitions are processed through target and predictor networks, the embedding discrepancy on an unlabeled dataset serves as an intrinsic reward to guide offline policy learning.

## 2 Related Work

### 2.1 Offline RL with Reward Annotations

Traditional offline RL assumes datasets include reward annotations, enabling algorithms such as Conservative Q-Learning (Kumar et al., 2020), Implicit Q-Learning (Kostrikov et al., 2022b), and Critic Regularized Regression (Wang et al., 2020) to optimize policies using supervised learning objectives (Levine et al., 2020; Fujimoto & Gu, 2021). However, crafting robust and generalizable reward functions often demands extensive domain expertise, posing a significant bottleneck. To address this, ORIL (Zolna et al., 2020) employs positive-unlabeled (PU) learning (Elkan & Noto, 2008) to infer rewards from reward-free datasets. While ReLOAD shares the goal of reward-free offline RL, it diverges by distilling rewards through embedding discrepancies from expert transitions, avoiding PU learning's reliance on binary classification.

### 2.2 Imitation Learning (IL)

Imitation Learning (IL) offers an alternative for policy learning without explicit rewards. Behavior Cloning (BC) (Pomerleau, 1988) treats policy learning as supervised regression but is prone to compounding errors from covariate shift. Inverse RL (IRL) methods, such as those by (Englert et al., 2013) and (Kostrikov et al., 2022a), infer reward functions to explain expert demonstrations, while adversarial IL, like GAIL (Ho & Ermon, 2016), uses a minimax game between a discriminator and policy. These approaches, however, can be unstable and sensitive to hyperparameters (Orsini et al., 2021). More recently, optimal transport-based IL methods (Xiao et al., 2019; Dadashi et al., 2022; Cohen et al., 2021) minimize Wasserstein distances (Villani & Villani, 2009) between expert and agent distributions, sidestepping adversarial instability. ReLOAD differs by using Random Network Distillation (RND) to infer rewards directly from expert embeddings, bypassing both supervised cloning and complex distribution matching.

### 2.3 Reward Inference in Offline RL

Reward inference is a growing focus in offline RL, especially where intrinsic motivation, well-explored in online RL (Brown et al., 2019; Ibarz et al., 2018; Yu et al., 2020), is adapted to static datasets. Unlike online methods that often leverage annotated data or environment interactions, ReLOAD targets fully reward-free and action-label-free offline reward inference settings. Recent approaches include Optimal Transport Reward (OTR) (Luo et al., 2023) labeling, which assigns rewards via Wasserstein-based couplings between agent and expert trajectories (Villani & Villani, 2009). On the other hand, Calibrated Latent Guidance (CLUE) (Liu et al., 2023) uses a conditional variational auto-encoder (VAE) (Kingma, 2013) to map expert and agent transitions into a shared latent space, computing rewards from Euclidean distances to a collapsed expert embedding. Further, Aligned Imitation Learning via Optimal Transport (ALIOT) (Bobrin et al., 2024) aligns transitions in a structured intent representation space, reducing cost function sensitivity of OTR.

However, these methods come with notable drawbacks. OTR requires careful selection of a cost function (e.g., cosine or Euclidean), which often demands environment-specific tuning. AILOT, while addressing some of OTR's limitations by operating in a latent intent space, still relies on solving OT problems—typically via the Sinkhorn algorithm—which scales poorly with dataset size and dimensionality. Moreover, AILOT also assumes access to sufficient unlabeled trajectories to extract intents. Furthermore, in the case of multimodality of intentions, its intent alignment may fail to capture diverse behaviors without additional clustering.

Similarly, CLUE's VAE-based approach struggles with such data unless supplemented with clustering, adding further complexity. CLUE also requires access to the action label to infer an intrinsic reward.

In contrast, ReLOAD employs RND to train a predictor network against a fixed traget network using transitions from $\mathcal{D}_e$ (samples of expert distribution $D_E$), then uses embedding discrepancies to quantify transition novelty across $\mathcal{D}_a$ (samples of a broad distribution $D_U$). This avoids the computational overhead of optimal transport, latent space modeling, and dependency on action labels, offering a lightweight and scalable solution.

## 3 Preliminaries

### 3.1 Offline RL

We consider a standard Reinforcement Learning (RL) problem defined by a Markov Decision Process (MDP) $\mathcal{M} = (\mathcal{S}, \mathcal{A}, p, r, \rho_0, \gamma)$, where: $\mathcal{S} \subset \mathbb{R}^n$ is the state space, $\mathcal{A} \subset \mathbb{R}^m$ is the action space, $p : \mathcal{S} \times \mathcal{A} \to \mathcal{P}(\mathcal{S})$ denotes the transition dynamics, $r : \mathcal{S} \times \mathcal{A} \to \mathbb{R}$ is the extrinsic reward function, $\rho_0$ is the initial state distribution, $\gamma \in (0, 1]$ is the discount factor. The goal in RL is to learn a policy $\pi_\phi(a|s)$ that maximizes the expected discounted return: $\mathbb{E}_{\tau \sim \pi_\phi} \left[ \sum_{t=0}^{\infty} \gamma^t r(s_t, a_t) \right]$. Where a trajectory $\tau = (s_0, a_0, s_1, a_1, \dots)$ is generated by iteratively sampling $s_0 \sim \rho_0$, $a_t \sim \pi_\phi(\cdot|s_t)$, and $s_{t+1} \sim p(\cdot|s_t, a_t)$.

In the offline RL setting, interaction with the environment is not permitted. Instead, the agent learns from fixed datasets without further exploration. We assume access to reward-free offline transitions $\mathcal{D}_a = \{(s_i, a_i, s_i')\}_{i=1}^N$, collected from various policies (e.g., a behavior policy or a mixture of policies), representing a broad distribution of transitions, which we denote as samples from $D_U$. Additionally, we are provided with a limited set of expert transitions, $\mathcal{D}_e = \{(s_i, s_i')\}_{i=1}^M$, collected from the same MDP $\mathcal{M}$, assumed to reflect high-quality behavior, and treated as samples from an expert distribution $D_E$. By omitting actions from $\mathcal{D}_e$, our method focuses on state transition patterns, ensuring applicability across different dynamics. The objective is to learn an offline policy that emulates expert performance by leveraging $\mathcal{D}_e$ (expert guidance) alongside $\mathcal{D}_a$ (broader context), without relying on extrinsic rewards.

### 3.2 Random Network Distillation (RND)

Random Network Distillation (RND) (Burda et al., 2018) is a technique originally developed for online RL to encourage exploration by quantifying novelty and later adapted for offline RL to guide imitation or anti-exploration (Nikulin et al., 2023). It compares the outputs of two networks: a fixed target network and a trainable predictor network.

In our formulation, RND operates on transitions $x = (s, s') \in \mathcal{S} \times \mathcal{S}$, reflecting state changes in the MDP. Formally:

- **Target Network**: $f_\psi : \mathcal{S} \times \mathcal{S} \to \mathbb{R}^K$, initialized with random weights $\psi$ and kept fixed.

- **Predictor Network**: $g_\theta : \mathcal{S} \times \mathcal{S} \to \mathbb{R}^K$, trained to approximate the target network's output.

The predictor is optimized over the expert distribution $D_E$ with the loss:

$$L_{\text{pred}}(\theta) = \mathbb{E}_{x \sim D_E} \left[ \| f_\psi(x) - g_\theta(x) \|_2^2 \right], \tag{1}$$

where gradient flow through $f_\psi$ is halted. Once trained on the expert distribution, the prediction error $\| f_\psi(x) - g_\theta(x) \|_2^2$ is small for transitions $x = (s, s')$ similar to those in $D_E$ (expert-like behavior) and large for transitions that deviate, such as those from $D_U$. In offline imitation learning, this error serves as an intrinsic reward signal, guiding the policy toward expert behavior by penalizing deviations from $D_E$. This approach provides a principled way to label transitions when extrinsic rewards are unavailable, aligning policy learning with expert demonstrations.

## 4 Formal Justification of the Novelty Reward

Our method repurposes Random Network Distillation (RND) for imitation learning, in contrast to its conventional use in exploration or novelty suppression. Prior work has applied RND in offline RL settings to penalize state-action transitions with high prediction error, thereby discouraging out-of-distribution behavior and encouraging conservative policies (Nikulin et al., 2023). In ReLOAD, however, RND is reinterpreted as a reward distillation mechanism: the predictor is trained exclusively on expert state transitions, and the resulting prediction error serves as a proxy for state transition quality, lower for expert-like behavior, and higher otherwise. This reframes the inductive bias from "uncertainty avoidance" to "expert alignment".

Moreover, ReLOAD provides a formal justification for this reward structure: as we show in Theorem 1, the expected prediction error is strictly lower under the expert distribution than under the broader offline dataset. This perspective has not been explored in prior RND-based offline RL methods and enables ReLOAD to recover expert-like behavior under minimal supervision, using a simple and generalizable framework. Below, we formalize this argument and present the theoretical basis of our reward mechanism.

**Theorem 1.** *Let $D_E$ and $D_U$ be probability distributions over state transitions $X$ (where $x = (s, s')$ represents a transition), and the support of $D_E$ is a proper subset of $D_U$, i.e., $\mathrm{supp}(D_E) \subsetneq \mathrm{supp}(D_U)$.*

*Define $f : X \to \mathbb{R}^K$ as a fixed, randomly initialized target network, and $g : X \to \mathbb{R}^K$ as a predictor network trained on samples from $D_E$ to minimize the objective:*

$$L_{pred}(\theta) = \mathbb{E}_{x \sim D_E} \|f(x) - g(x)\|_2^2, \tag{2}$$

*Let the expected prediction error on $D_E$ be:*

$$\mu_E = \mathbb{E}_{x \sim D_E} \|f(x) - g(x)\|_2^2. \tag{3}$$

*Assume that:*

1. *The distribution $D_U$ and $D_E$ agree on $\mathrm{supp}(D_E)$, i.e.,*

$$\mathbb{E}_{x \sim D_U} [\|f(x) - g(x)\|_2^2 \mid x \in \mathrm{supp}(D_E)] = \mu_E. \tag{4}$$

2. *There exists a measurable subset $S \subset \mathrm{supp}(D_U) \setminus \mathrm{supp}(D_E)$ with $\Pr_{x \sim D_U}(x \in S) > 0$, such that the expected prediction error on $S$ is strictly greater than $\mu_E$:*

$$\mathbb{E}_{x \sim D_U} [\|f(x) - g(x)\|_2^2 \mid x \in S] > \mu_E. \tag{5}$$

*Then, the expected prediction error over $D_U$ exceeds that over $D_E$:*

$$\mathbb{E}_{x \sim D_U} \|f(x) - g(x)\|_2^2 > \mu_E. \tag{6}$$

**Proof:** To prove the claim, We decompose the expectation over $D_U$ into two regions:

$$\mathbb{E}_{x \sim D_U} \|f(x) - g(x)\|_2^2 = \int_{\mathrm{supp}(D_E)} \|f(x) - g(x)\|_2^2 dD_U(x) + $$
$$\int_{\mathrm{supp}(D_U) \setminus \mathrm{supp}(D_E)} \|f(x) - g(x)\|_2^2 dD_U(x). \tag{7}$$

Define $p = \Pr_{x \sim D_U}(x \in \mathrm{supp}(D_E))$ as the probability that a sample from $D_U$ lies in the support of $D_E$. Since $\mathrm{supp}(D_E) \subsetneq \mathrm{supp}(D_U)$, and assuming $D_U$ assigns positive probability to $\mathrm{supp}(D_U) \setminus \mathrm{supp}(D_E)$, we have $0 \le p < 1$.

Using conditional expectations, rewrite the total expectation as:

$$\mathbb{E}_{x \sim D_U} \|f(x) - g(x)\|_2^2 = p \cdot \mathbb{E}_{x \sim D_U} [\|f(x) - g(x)\|_2^2 \mid x \in \mathrm{supp}(D_E)] + $$
$$(1 - p) \cdot \mathbb{E}_{x \sim D_U} [\|f(x) - g(x)\|_2^2 \mid x \notin \mathrm{supp}(D_E)]. \tag{8}$$

By assumption (1), we have:

$$\mathbb{E}_{x \sim D_U}[\|f(x) - g(x)\|_2^2 \mid x \in \mathrm{supp}(D_E)] = \mu_E. \tag{9}$$

Thus, the first term satisfies:

$$p \cdot \mathbb{E}_{x \sim D_U}[\|f(x) - g(x)\|_2^2 \mid x \in \mathrm{supp}(D_E)] = p \cdot \mu_E. \tag{10}$$

For the second region $\mathrm{supp}(D_U) \setminus \mathrm{supp}(D_E)$, consider the expected error:

$$\mathbb{E}_{x \sim D_U}[\|f(x) - g(x)\|_2^2 \mid x \notin \mathrm{supp}(D_E)]. \tag{11}$$

By assumption (2), there exists a subset $S \subset \mathrm{supp}(D_U)$ with $\Pr_{x \sim D_U}(s \in S) > 0$ and $\mathbb{E}_{x \sim D_U}[\|f(x) - g(x)\|_2^2 \mid x \in S] > \mu_E$.

Since $S$ is a subset of $\mathrm{supp}(D_U) \setminus \mathrm{supp}(D_E)$, and the error on the remaining parts of $\mathrm{supp}(D_U) \setminus \mathrm{supp}(D_E)$ is non-negative, the conditional expectation over the entire complement satisfies:

$$\mathbb{E}_{x \sim D_U}[\|f(x) - g(x)\|_2^2] \mid x \notin \mathrm{supp}(D_E)] \geq \mathbb{E}_{x \sim D_U}[\|f(x) - g(x)\|_2^2] \mid x \in \mathcal{S}] > \mu_E. \tag{12}$$

Thus, the second term is:

$$(1 - p) \cdot \mathbb{E}_{x \sim D_U}[\|f(x) - g(x)\|_2^2 \mid x \notin \mathrm{supp}(D_E)] > (1 - p) \cdot \mu_E, \tag{13}$$

with $1 - p > 0$ because $\mathrm{supp}(D_E) \subsetneq \mathrm{supp}(D_U)$.

Combining the two terms, we derive:

$$\mathbb{E}_{x \sim D_U}\|f(x) - g(x)\|_2^2 = p \cdot \mu_E + (1 - p) \cdot \mathbb{E}_{x \sim D_U}[\|f(x) - g(x)\|_2^2 \mid x \notin \mathrm{supp}(D_E)]. \tag{14}$$

Since $\mathbb{E}_{x \sim D_U}[\|g(x) - f(x)\|_2^2] \mid x \notin \mathrm{supp}(D_E)] > \mu_E,$, it follows that:

$$\mathbb{E}_{x \sim D_U}\|f(x) - g(x)\|_2^2 > p \cdot \mu_E + (1 - p) \cdot \mu_E = \mu_E. \tag{15}$$

This strict inequality holds because the second term contributes the excess over $\mu_E$ with positive weight $1 - p$.

Thus, we conclude:

$$\mathbb{E}_{x \sim D_U}\|f(x) - g(x)\|_2^2 > \mu_E. \tag{16}$$

This result implies that if we define a reward function $r(x) = -\|f(x) - g(x)\|_2^2$, transitions from $D_E$ (expert-like, low error) yield higher rewards, while those unique to $D_U$ (novel states, high error) yield lower rewards, which align with the goals of Imitation learning.

Further, training the predictor solely on expert demonstrations induces a behavior prior. The optimization implicitly encourages policies that minimize divergence from expert-like behaviors because the learned reward function assigns higher values to expert-aligned transitions.

Unlike Inverse RL, which explicitly optimizes a reward function, ReLOAD self-supervises reward labeling. Since the predictor never trains on non-expert transitions, it avoids reward contamination, ensuring robustness against noisy, mixed-quality datasets.

Also, unlike AILOT, which requires mapping states to an intent space and solving optimal transport problems—a computationally intensive process sensitive to the quality of intent representations—ReLOAD's reward computation involves only forward passes through neural networks after initial training. This simplicity reduces computational demands and enhances scalability, making ReLOAD particularly well-suited for large, diverse offline datasets.

---

**Algorithm 1** Self-Supervised Reward Annotation via RND

---

1: **Inputs:** Offline dataset $\mathcal{D} = \{(s, a, s')\}$, Expert transitions $\mathcal{D}_e = \{(s, s')\}$, Target network $f_\psi$, Predictor network $g_\theta$, Learning rate $\eta$, Number of epochs $N_{\text{pred}}$, Hyperparameters $\alpha, \beta$

2: **Initialize:** Fixed target network $f_\psi$ with parameters $\psi$, Trainable predictor network $g_\theta$ with parameters $\theta$

3: **Step 1: Pretraining the Predictor with RND**

4: **for** epoch $= 1$ to $N_{\text{pred}}$ **do**

5:     Sample $(s, s')$ from $\mathcal{D}_e$

6:     Compute RND loss:
$$L_{\text{pred}}(\theta) = \|f_\psi(s, s') - g_\theta(s, s')\|_2^2$$

7:     Update predictor: $\theta \leftarrow \theta - \eta \nabla_\theta L_{\text{pred}}(\theta)$

8: **end for**

9: **Step 2: Computing Reward for Offline Data**

10: **for** $(s, a, s') \in \mathcal{D}$ **do**

11:     Compute intrinsic reward:
$$r_{\text{RND}}(s, s') = -\|f_\psi(s, s') - g_\theta(s, s')\|_2^2$$

12:     Optionally, apply exponential transformation:
$$\tilde{r}(s, s') = \alpha \exp(\beta r_{\text{RND}}(s, s'))$$

13: **end for**

14: **Output:** Labeled dataset $\tilde{\mathcal{D}} = \{(s, a, \tilde{r}(s, s'), s')\}$ or $\{(s, a, r_{\text{RND}}(s, s'), s')\}$

---

## 5 Method

Our approach adapts Random Network Distillation (RND) to derive an intrinsic reward signal from expert transitions for offline imitation learning. It consists of three stages: (i) aligning a predictor network with a fixed target network using expert transitions, (ii) assigning intrinsic rewards to an unlabeled dataset based on embedding differences, and (iii) training an offline policy with these rewards to mimic expert behavior. This eliminates the need for manually defined rewards in offline reinforcement learning.

**Expert Embedding Alignment:** We start with an expert dataset $\mathcal{D}_e = \{(s_i, s_i')\}_{i=1}^M$, where each $(s_i, s_i')$ is a state transition from expert demonstrations, assumed to be sampled from an expert distribution $D_E$. We define two neural networks:

- **Target Network**: $f_\psi : \mathcal{S} \times \mathcal{S} \to \mathbb{R}^K$, a fixed, randomly initialized network with parameters $\psi$, mapping transitions to a $K$-dimensional embedding space.

- **Predictor Network**: $g_\theta : \mathcal{S} \times \mathcal{S} \to \mathbb{R}^K$, a trainable network with parameters $\theta$, mapping transitions to the same embedding space.

The predictor $g_\theta$ is trained on a batch $M$ to match the target network's outputs on expert transitions by minimizing:

$$L_{\text{pred}}(\theta) = \frac{1}{M} \sum_{i=1}^M \|f_\psi(s_i, s_i') - g_\theta(s_i, s_i')\|_2^2. \tag{17}$$

This loss approximates $\mathbb{E}_{x \sim D_E}\|f_\psi(x) - g_\theta(x)\|_2^2$, where $x = (s, s')$, ensuring $g_\theta$ learns embeddings aligned with expert behavior while $f_\psi$ remains a stable target.

**Intrinsic Reward Assignment via Embedding Discrepancy:** We then apply the trained predictor to an unlabeled offline dataset $\mathcal{D}_a = \{(s_j, s'_j)\}_{j=1}^N$, assumed to be sampled from a broader distribution $D_U$. The change in subscript from $i$ in equation 17 to $j$ here is intentional: the predictor $f$ is trained on expert transitions $(s_i, s'_i) \in D_E$, and then evaluated on unlabeled transitions $(s_j, s'_j) \in \mathcal{D}_a$. For notational convenience, we denote each transition as $x_j = (s_j, s'_j)$. The intrinsic reward assigned to a transition $x_j$ is then given by:

$$r_{\text{RND}}(x_j) = -\|f_\psi(x_j) - g_\theta(x_j)\|_2^2. \tag{18}$$

This reward is higher (less negative) for transitions similar to those in $\mathcal{D}_e$, where $g_\theta$ closely matches $f_\psi$, and lower (more negative) for dissimilar transitions, where discrepancies are larger. This reflects the theorem's insight that $\mathbb{E}_{x \sim D_U}\|f_\psi(x) - g_\theta(x)\|_2^2 > \mathbb{E}_{x \sim D_E}\|f_\psi(x) - g_\theta(x)\|_2^2$, prioritizing expert-like behavior.

**Offline Policy Learning with Distilled Rewards:** Using these rewards, we create a labeled dataset:

$$\tilde{D} = \{(s_j, r_{\text{RND}}(x_j), s'_j)\}_{j=1}^N. \tag{19}$$

A policy $\pi_\phi$, parameterized by $\phi$, is trained using an offline RL algorithm (e.g., Implicit Q-Learning (Kostrikov et al., 2021)) to maximize the expected return as estimated by the Q-function learned from the offline dataset, which is formulated as:

$$\max_\phi \ \mathbb{E}_{\tilde{D}} \left[ \hat{Q}^\pi(s, a) \right], \tag{20}$$

where $\hat{Q}^\pi$ is estimated from $\tilde{D}$ with rewards $r_{\text{RND}}$, and actions $a$ are derived from transitions assuming a deterministic environment or dataset-provided actions.

**Reward Squashing for Stability:** To improve training stability, we optionally transform rewards as:

$$\tilde{r}(x_j) = \alpha \exp(\beta r_{\text{RND}}(x_j)), \tag{21}$$

where $\alpha > 0$ scales the reward, and $\beta > 0$ adjusts sensitivity. Since $r_{\text{RND}}(x_j) \leq 0$, this maps rewards to $(0, \alpha]$, bounding variance and aiding convergence.

In conclusion, ReLOAD's intrinsic reward mechanism is simple and theoretically grounded. By training the predictor network $g_\theta$ exclusively on expert transitions from $\mathcal{D}_e$, we ensure that the prediction error $\|f_\psi(x) - g_\theta(x)\|_2^2$ is minimized for expert-like behavior and maximized for suboptimal or novel transitions. This is formally supported by Theorem 1, which proves that the expected prediction error over the broader dataset $\mathcal{D}_a$ exceeds that over the expert dataset $\mathcal{D}_e$, providing a clean and reliable reward signal. Unlike AILOT, which requires mapping states to an intent space and solving optimal transport problems—a computationally intensive process sensitive to the quality of intent representations—ReLOAD's reward computation involves only forward passes through neural networks after initial training. This simplicity reduces computational demands and enhances scalability, making ReLOAD well-suited for large, diverse offline datasets.

## 6 Experiments

We evaluate our method across a range of benchmark tasks to address the following key research questions:

- Can our method enable offline RL algorithms to recover performance comparable to those using well-engineered ground-truth rewards?

- How effectively does our novelty reward formulation leverage expert demonstrations to enhance policy learning?

Table 1: Performance Comparison on D4RL Locomotion Tasks. The reported results are normalized scores (mean ± standard deviation) across 10 random seeds. For these methods, the results are given for K = 1, the number of expert trajectories. The highest scores are highlighted in green.

| Dataset | IQ-Learn | SQIL | ORIL | SMODICE | ReLOAD |
|---|---|---|---|---|---|
| halfcheetah-medium | 21.7 ± 1.5 | 24.3 ± 2.7 | 56.8 ± 1.2 | 42.4 ± 0.6 | 48.2 ± 0.7 |
| halfcheetah-medium-replay | 6.7 ± 1.8 | 43.8 ± 1.0 | 46.2 ± 1.1 | 38.3 ± 2.0 | 42.6 ± 0.8 |
| halfcheetah-medium-expert | 2.0 ± 0.4 | 6.7 ± 1.2 | 48.7 ± 2.4 | 81.0 ± 2.3 | 92.6 ± 2.5 |
| hopper-medium | 29.6 ± 5.2 | 66.9 ± 5.1 | 96.3 ± 0.9 | 54.8 ± 1.2 | 78.7 ± 6.9 |
| hopper-medium-replay | 23.0 ± 9.4 | 98.6 ± 0.7 | 56.7 ± 12.9 | 30.4 ± 1.2 | 95.9 ± 2.2 |
| hopper-medium-expert | 9.1 ± 2.2 | 13.6 ± 9.6 | 25.1 ± 12.8 | 82.4 ± 7.7 | 104.8 ± 9.0 |
| walker2d-medium | 5.7 ± 4.0 | 51.9 ± 11.7 | 20.4 ± 13.6 | 67.8 ± 6.0 | 81.3 ± 1.0 |
| walker2d-medium-replay | 17.0 ± 7.6 | 42.3 ± 5.8 | 71.8 ± 9.6 | 49.7 ± 4.6 | 78.6 ± 2.5 |
| walker2d-medium-expert | 7.7 ± 2.4 | 18.8 ± 13.1 | 11.6 ± 14.7 | 94.8 ± 11.1 | 110.5 ± 0.7 |
| D4RL Locomotion total | 122.5 | 366.9 | 433.6 | 541.6 | 733.2 |

| Dataset | IQL | OTR | CLUE | AILOT | ReLOAD |
|---|---|---|---|---|---|
| halfcheetah-medium | 47.4 ± 0.2 | 43.3 ± 0.2 | 45.6 ± 0.3 | 47.7 ± 0.35 | 48.2 ± 0.7 |
| halfcheetah-medium-replay | 44.2 ± 1.2 | 41.3 ± 0.6 | 43.5 ± 0.5 | 42.4 ± 0.8 | 42.6 ± 0.8 |
| halfcheetah-medium-expert | 86.7 ± 5.3 | 89.6 ± 3.0 | 91.4 ± 2.1 | 92.4 ± 1.54 | 92.6 ± 2.5 |
| hopper-medium | 66.2 ± 5.7 | 78.7 ± 5.5 | 78.3 ± 5.4 | 82.2 ± 5.6 | 78.7 ± 6.9 |
| hopper-medium-replay | 94.7 ± 8.6 | 84.8 ± 2.6 | 94.3 ± 6.0 | 98.7 ± 0.4 | 95.9 ± 2.2 |
| hopper-medium-expert | 91.5 ± 14.3 | 93.2 ± 20.6 | 96.5 ± 14.7 | 103.4 ± 5.3 | 104.8 ± 9.0 |
| walker2d-medium | 78.3 ± 8.7 | 79.4 ± 1.4 | 80.7 ± 1.5 | 78.3 ± 0.8 | 81.3 ± 1.0 |
| walker2d-medium-replay | 73.8 ± 7.1 | 66.0 ± 6.7 | 76.3 ± 2.8 | 77.5 ± 3.1 | 78.6 ± 2.5 |
| walker2d-medium-expert | 109.6 ± 1.0 | 109.3 ± 0.8 | 109.3 ± 2.1 | 110.2 ± 1.2 | 110.5 ± 0.7 |
| D4RL Locomotion total | 692.4 | 685.6 | 714.5 | 732.8 | 733.2 |

- How does our method perform in fully offline imitation learning compared to state-of-the-art offline RL approaches?

- What is the impact of varying the number of expert demonstrations on performance?

We conduct extensive experiments on standard offline RL and offline IL benchmarks to answer these questions, including D4RL Locomotion, Antmaze, and Adroit.

A key challenge in offline RL is handling datasets that contain a mixture of expert and suboptimal behaviors. We systematically evaluate our method on datasets with varying levels of expert supervision, analyzing its ability to identify and propagate useful reward signals. To further assess robustness, we conduct an ablation study by varying the number of high-quality demonstrations and examining the impact on learning performance.

All experiments were conducted with over 10 random seeds, with performance measured using average episode return and normalized scores relative to ground-truth reward baselines. We also report runtime complexity to assess computational efficiency. Our results demonstrate that our self-supervised reward formulation significantly enhances offline RL performance while providing a strong alternative to manually engineered reward functions in offline imitation learning settings.

Table 2: Performance Comparison on sparse AntMaze Tasks. The reported results are normalized scores across 10 seeds. The reposted results are given for $K = 1$ number of expert trajectories. The ReLOAD approach shows the most consistent performance, while AILOT achieves a slightly higher total reward.

| Dataset | IQL | OTR | CLUE | AILOT | ReLOAD |
|---|---|---|---|---|---|
| antmaze-umaze | 88.7 | $81.6 \pm 7.3$ | $92.1 \pm 3.9$ | $93.5 \pm 4.8$ | $95 \pm 5.4$ |
| antmaze-umaze-diverse | 67.5 | $70.4 \pm 8.9$ | $68.0 \pm 11.2$ | $63.4 \pm 7.6$ | $67.5 \pm 14.5$ |
| antmaze-medium-play | 72.9 | $73.9 \pm 6.0$ | $75.3 \pm 6.3$ | $71.3 \pm 5.2$ | $72.9 \pm 12.5$ |
| antmaze-medium-diverse | 72.1 | $72.5 \pm 6.9$ | $74.6 \pm 7.5$ | $75.5 \pm 7.4$ | $76.3 \pm 14.1$ |
| antmaze-large-play | 43.2 | $49.7 \pm 6.9$ | $55.8 \pm 7.7$ | $57.6 \pm 6.6$ | $61.6 \pm 12.0$ |
| antmaze-large-diverse | 46.9 | $48.1 \pm 7.9$ | $49.9 \pm 6.9$ | $66.6 \pm 3.1$ | $51.4 \pm 14.2$ |
| AntMaze total | 391.3 | 396.2 | 415.7 | 427.9 | 424.7 |

## 6.1 Implementation Details

Our Random Network Distilled Reward learning module can be plugged into any offline RL algorithm. For our implementation, we use Implicit-Q Learning (IQL) (Kostrikov et al., 2021), which aligns with other baselines that also build their approach on this offline RL algorithm. We implement ReLOAD in JAX (Bradbury et al., 2018) and use official IQL implementation[1]. We also test our approach against the Behavior Cloned (BC) policy. The details of all the hyperparameters and network architecture are in the Appendix B.

## 6.2 Runtime

All of the experiments were conducted on a single NVIDIA A40 GPU. The runtime overhead of ReLOAD is less than a minute (compared to the offline RL algorithm training time), including training the predictor network and annotating offline data. Thus, the total runtime of ReLOAD+IQL equals approximately 20 minutes. We use 100 training iterations with a batch size 64 to train the RND predictor network. Moreover, unlike previous approaches such as OTR, where computational cost increases with dimensionality, ReLOAD remains mostly invariant to the dimensionality of the state space. For high-dimensional tasks (e.g., learning from pixels), ReLOAD is more suitable as the reward annotation depends on the embedding generated from the predictor and target network. Compared to ReLOAD, AILOT takes approximately 10 minutes overhead to train the latent space and reward relabeling.

## 6.3 Baselines

We compare **ReLOAD** with the following baselines (refer Appendix C for description of additional baselines):

- **IQL** (Kostrikov et al., 2021) is a leading offline RL algorithm that mitigates out-of-distribution action selection by interpreting the value function as a random variable. The upper bound on uncertainty is controlled via expectile regression. For evaluation, we use the ground-truth reward from D4RL tasks.

- **OTR** (Luo et al., 2023) defines a reward function by leveraging the optimal transport distance between state distributions from expert demonstrations and an unlabeled offline dataset.

- **CLUE** (Liu et al., 2023) constructs a variational autoencoder (VAE) (Kingma, 2013) for calibrating the latent space of expert and agent state-action transitions. The intrinsic reward is computed as the discrepancy between the agent's transitions and the averaged expert representation.

---

[1]https://github.com/ikostrikov/implicit_q_learning

Table 3: Performance Comparison on Adroit Tasks. The reported results are normalized scores across 10 seeds. The reposted results are given for $K = 1$ number of expert trajectories. The ReLOAD approach shows the competitive performance with the current state-of-the-art method AILOT.

| Dataset | IQL | OTR | CLUE | AILOT | ReLOAD |
|---|---|---|---|---|---|
| door-cloned | 1.6 | $0.01 \pm 0.01$ | $0.02 \pm 0.01$ | $0.05 \pm 0.02$ | $1.7 \pm 2.9$ |
| door-human | 4.3 | $5.9 \pm 2.7$ | $7.7 \pm 3.9$ | $7.9 \pm 3.2$ | $7.7 \pm 1.6$ |
| hammer-cloned | 2.1 | $0.9 \pm 0.3$ | $1.4 \pm 1.0$ | $1.6 \pm 0.1$ | $1.8 \pm 1.4$ |
| hammer-human | 1.4 | $1.8 \pm 1.4$ | $1.9 \pm 1.2$ | $1.8 \pm 1.3$ | $1.8 \pm 1.1$ |
| pen-cloned | 37.3 | $46.9 \pm 20.9$ | $59.4 \pm 21.1$ | $61.4 \pm 19.5$ | $64.5 \pm 24.6$ |
| pen-human | 71.5 | $66.8 \pm 21.2$ | $82.9 \pm 20.2$ | $89.4 \pm 0.1$ | $88.3 \pm 8.7$ |
| relocate-cloned | -0.2 | $-0.24 \pm 0.03$ | $-0.23 \pm 0.02$ | $-0.20 \pm 0.03$ | $-0.11 \pm 0.04$ |
| relocate-human | 0.1 | $0.1 \pm 0.1$ | $0.2 \pm 0.3$ | $0.28 \pm 0.1$ | $0.23 \pm 0.2$ |
| Adroit total | 118.1 | 122.2 | 153.3 | 162.2 | 165.9 |

- **AILOT** (Bobrin et al., 2024) define a reward function that uses optimal transport between the special representations of states in the form of intents that incorporate pairwise spatial distances within the data.

## 6.4 Results

We evaluate ReLOAD across a range of D4RL locomotion tasks using IQL as the fixed offline RL backbone. This ensures a consistent comparison across all experiments. Notably, several baselines such as OTR, CLUE, and AILOT also use IQL, making our comparisons fair and aligned. All experiments were conducted with over 10 random seeds, with performance reported as normalized scores (or returns) alongside standard deviations. The best-rewarding trajectory is selected from the offline dataset as the expert trajectory. All the reward and action annotations are discarded for forward inference, and trajectories are relabeled based on intrinsic rewards. We compare ReLOAD against competitive baselines, including IQ-Learn, SQIL, ORIL, SMODICE, OTR, CLUE, and AILOT.

On D4RL locomotion tasks, ReLOAD achieves strong performance, Table 1, outperforming several reward-free baselines under minimal supervision (e.g., K=1 expert trajectory), without relying on action labels (as required by CLUE) or costly intent alignment via optimal transport (as in AILOT). ReLOAD also performs superior to IQL with true reward-annotated trajectories. ReLOAD matches or exceeds prior reward annotation methods on 7 out of 9 locomotion benchmarks, highlighting its ability to infer effective reward signals directly from state transitions.

In sparse-reward settings such as AntMaze and Adroit, ReLOAD demonstrates robust generalization and competitive performance relative to alignment-based approaches. For the AntMaze task, we add a reward bias of $-3$ to align with the binary reward nature of true reward annotation, which is in line with other reward annotation approaches. Although AILOT outperforms ReLOAD in AntMaze-large-diverse variant, ReLOAD remains highly competitive while avoiding the computational overhead of latent space modeling. Although AILOT achieves the highest cumulative reward, ReLOAD performs best on most AntMaze navigation tasks, outperforming all baselines as shown in Table 2.

In the Adroit Manipulation task, Table 3, ReLOAD performs better than all the baselines. Further, in more ambiguous settings—such as door-cloned and relocate-cloned—methods like OTR, CLUE, and AILOT fail to recover informative reward signals, likely due to the presence of mixed-quality demonstrations and

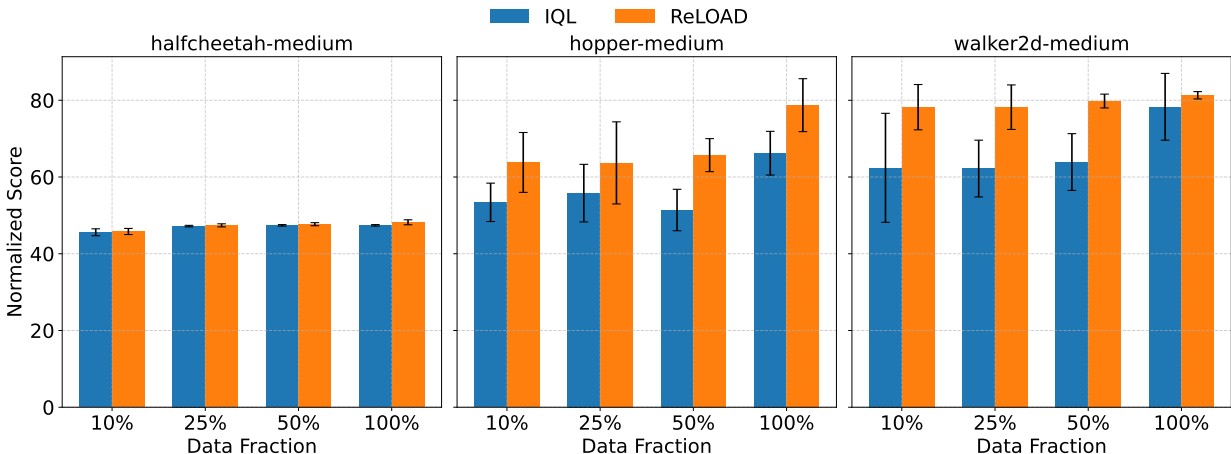

Figure 2: Comparison of IQL and ReLOAD on D4RL Locomotion Tasks across varying fractions of offline data (10%, 25%, 50%, 100%). ReLOAD consistently outperforms IQL, particularly in Hopper and Walker2d environments. Bars indicate mean normalized scores with standard deviation over 10 random seeds.

human-induced noise. Specifically, we believe AILOT fails due to difficulty in extracting meaningful intent from noisy offline data similar. In contrast, ReLOAD continues to perform reliably, illustrating its robustness to imperfect expert data. These results suggest that ReLOAD's simple embedding-based reward mechanism is well-suited for clean and noisy offline RL datasets without explicit supervision or complex alignment procedures.

### 6.5 Effect of Number of Expert Trajectories

In this subsection, we investigate the sensitivity of ReLOAD to the number of expert trajectories available in the expert dataset $\mathcal{D}_e$, comparing its performance against other baselines. Table 13 reports results for K=1 and K=5 expert demonstrations across D4RL locomotion tasks. Even with a single trajectory, ReLOAD achieves strong performance, highlighting its ability to generalize with minimal supervision. Increasing the number of demonstrations leads to consistent, though moderate, improvements, most notably in the Hopper and Walker2d environments, indicating that additional expert guidance refines the reward signal without requiring architectural or algorithmic changes.

These findings suggest two important properties of ReLOAD. First, it is highly data-efficient: a single expert trajectory suffices to recover competitive behavior across a wide range of tasks. Second, the RND-based reward formulation gracefully scales with the quality and quantity of supervision, enabling smooth reward shaping as more demonstrations become available. Importantly, this is achieved without complex modeling of expert intent, latent alignment procedures, or reliance on action labels—distinguishing ReLOAD from methods that require extensive preprocessing or structured supervision. This makes the approach particularly suitable for applications with varying levels of expert access and constrained annotation budgets.

### 6.6 Performance of ReLOAD Under Varying Offline Data Sizes

We compare the data efficiency of ReLOAD and IQL by evaluating their performance on offline datasets of different sizes. ReLOAD utilizes intrinsic rewards generated from expert demonstrations, whereas IQL relies on true environment rewards. Our experiments focus on three D4RL Locomotion tasks: halfcheetah-medium, hopper-medium, and walker2d-medium. For each task, we created subsets of the offline dataset at various proportions of the full data. Both ReLOAD and IQL were trained on these subsets, and their performance was assessed using normalized scores averaged over 10 random seeds. As illustrated in Figure 2 and Table 12, ReLOAD consistently outperforms IQL across all tasks and dataset sizes.

These results highlight the benefits of ReLOAD's intrinsic reward mechanism, which provides a more informative learning signal compared to the true rewards used by IQL. These findings suggest that reward annotation generated by ReLOAD propagates to efficient learning even with a low proportion of offline data. This enhanced data efficiency is particularly valuable in real-world scenarios where collecting large-scale offline datasets can be costly or impractical. The experiments demonstrate that ReLOAD can learn effective policies with significantly less data than traditional reward-based methods, making it a promising approach for diverse offline reinforcement learning applications.

## 7 Conclusion

We introduced ReLOAD, a reward-free offline reinforcement learning framework that distills intrinsic rewards from expert transitions using Random Network Distillation. This reframing of RND, from novelty avoidance to expert reward inference, highlights ReLOAD's conceptual departure from prior anti-exploration work. ReLOAD provides a simple yet effective mechanism to differentiate expert-like from suboptimal behaviors without relying on action labels or handcrafted rewards by measuring embedding discrepancies between a fixed prior and a trained predictor. Extensive evaluations across D4RL locomotion, AntMaze, and Adroit tasks show that ReLOAD matches or outperforms existing reward-free methods while being computationally lightweight and robust to limited expert supervision. These results highlight the potential of embedding-based reward inference as a scalable alternative for real-world offline RL applications.

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

## A    Implementation Details of Reward Labeling Module-RND

This section discusses the details of implementing the reward labeling strategy. We maintain two networks of identical architecture: the Target and the Predictor Network. The weights of the target networks are kept fixed. First, we extract the highest reward trajectory from the offline data based on the annotation provided in the offline data. The transitions (s, s') are concatenated and passed through the target and predictor networks. The predictor network is trained to minimize the mean square error loss between the output embedding of the target and the predictor network.

For offline data reward annotation, we freeze the weights of both target and predictor networks. The state transition from offline data is concatenated and passed through the target and predictor networks. The dissimilarity score (MSE) between target and predictor networks embedding, followed by reward squashing, is used to annotate offline data. The true reward annotation of the offline data is discarded if available.

This section discusses the details of implementing the reward labeling strategy. We maintain two networks of identical architecture: the Target and the Predictor Network. The weights of the target networks are kept fixed. First, we extract the highest reward trajectory from the offline data based on the annotation provided in the offline data. The transitions $(s, s')$ are concatenated and passed through the target and predictor networks. The predictor network is trained to minimize the mean square error loss between the output embedding of the target and the predictor network. We initialize the target network independently using standard Xavier initialization. The target network is frozen after initialization and never updated, while for the predictor network, we don't use any initialization strategy. We use Layer-Norm in both predictor and target networks.

For offline data reward annotation, we freeze the weights of both target and predictor networks. The state transition from offline data is concatenated and passed through the target and predictor networks. The dissimilarity score (MSE) between target and predictor networks embedding, followed by reward squashing, is used to annotate offline data. The true reward annotation of the offline data is discarded if available.

Table 4: Hyperparameters for training RND.

| Setting | MuJoCo and Antmaze | Adroit |
|---|:---:|:---:|
| Hidden dim | 256 | 256 |
| Number of layers | 2 | 2 |
| Batch size | 32 | 16 |
| Number of iterations | $10^2$ | 10 |
| Learning rate | $1e^{-3}$ | $3e^{-4}$ |
| Number of expert trajectories | 1 | 1 |

## B    Hyperparameters

The performance of ReLOAD is influenced by several key hyperparameters, which we detail below. These parameters were selected based on a combination of preliminary experiments and established practices in the offline reinforcement learning (RL) literature. The offline RL algorithm used is Implicit Q-Learning (IQL), and its hyperparameters are set according to the recommendations in the original IQL paper (Kostrikov et al., 2021).

Table 5 presents the hyperparameters used across all experiments unless otherwise specified. For certain tasks, minor adjustments were made to optimize performance; these parameters are highlighted in 6.

We adjusted the reward squashing parameters for specific tasks, such as the AntMaze and Adroit environments, to suit those domains' reward scales better. Specifically, for Locomotion task we use $\alpha = 10$ and $\beta = 5$ except for hopper-expert we use $\alpha = 10$ and $\beta = 0.5$, for AntMaze, we set $\alpha = 10$ and $\beta = 1$, while for Adroit, we used $\alpha = 10$ and $\beta = 5$. These adjustments were made based on initial experiments to ensure

Table 5: Hyperparameters used in the ReLOAD framework. These values were used across all experiments.

| Hyperparameter | Value | Description |
|---|---|---|
| Offline RL Algorithm | IQL | Choice of offline RL algorithm |
| Policy Learning Rate | 3e-4 | Learning rate for the policy network |
| Critic Learning Rate | 3e-4 | Learning rate for the critic network |
| Value Learning Rate | 3e-4 | Learning rate for value network |
| Discount Factor ($\gamma$) | 0.99 | Discount factor for future rewards |
| Training Steps | 1e6 | Number of training steps for IQL |
| Batch Size | 256 | Batch size for IQL training |

Table 6: Task dependent hyperparameters

| Task | Expectile | Temperature | $\alpha$ | $\beta$ |
|---|---|---|---|---|
| Locomotion | 0.7 | 6 | 10 | 5 |
| AntMaze | 0.9 | 10 | 10 | 1 |
| Adroit | 0.8 | 0.5 | 10 | 5 |

stable training and are consistent with practices in the offline RL community for handling varying reward magnitudes.

All the reported results are across 10 random seeds with 10 evaluation episodes for each seed, consistent with previous works. We also use reward scaling, which is similar to IQL. The reward scaling factor used is $\frac{1000}{\text{max\_return - min\_retrun}}$. For the Antmaze task, we also add a reward bias of $-3$ to the scaled reward.

## C    Baselines:

- **IQ-Learn** (Garg et al., 2021) is an imitation learning approach that implicitly embeds expert policy and rewards into an inverse Q-function, enabling direct policy learning from expert demonstrations.

- **Diffusion-QL** (Wang et al., 2022) reformulates policy learning as a conditional diffusion process, improving expressiveness and providing a more flexible regularization mechanism for aligning with the behavior policy that generated the dataset.

- **SQIL** (Reddy et al., 2019) introduces a soft Q-learning framework that assigns a fixed reward of one to expert transitions and zero to non-expert ones, effectively shaping the policy towards expert-like behavior.

- **ORIL** (Zolna et al., 2020) employs a discriminator to distinguish between optimal and suboptimal data in mixed datasets. The learned discriminator is then used for reward relabeling, guiding the policy toward expert-like behaviors.

- **SMODICE** (Ma et al., 2022) is an offline imitation learning method that minimizes the divergence between the state occupancy distributions of the agent and the expert. It achieves this through a dual formulation of the value function, allowing for direct state-matching IL without requiring explicit action supervision.

## D    Effect of Integrating ReLOAD with OTR

The integration of ReLOAD with Optimal Transport Reward (OTR) aims to explore whether combining RND-based embedding discrepancies with distributional alignment via optimal transport (OT) can enhance

Table 7: Effect of combining ReLOAd with OTR. The reported results are normalized scores across 10 seeds. The reposted results are given for $K = 1$ number of expert trajectories.

| Dataset | IQL | OTR | ReLOAD | ReLOAD+OTR |
|---------|-----|-----|--------|------------|
| antmaze-umaze | 88.7 | 81.6 ± 7.3 | 95 ± 5.4 | 90 ± 7.6 |
| antmaze-umaze-diverse | 67.5 | 70.4 ± 8.9 | 67.5 ± 14.5 | 65.7 ± 16.2 |
| antmaze-medium-play | 72.9 | 73.9 ± 6.0 | 72.9 ± 12.5 | 73.8 ± 11.9 |
| antmaze-medium-diverse | 72.1 | 72.5 ± 6.9 | 76.3 ± 14.1 | 75.0 ± 18.6 |
| antmaze-large-play | 43.2 | 49.7 ± 6.9 | 61.6 ± 12.0 | 58.0 ± 11.9 |
| antmaze-large-diverse | 46.9 | 48.1 ± 7.9 | 51.4 ± 14.2 | 51.7 ± 14.7 |
| AntMaze total | 391.3 | 396.2 | 424.7 | 414.2 |

reward inference. While ReLOAD leverages the simplicity of mean squared error (MSE) between fixed prior and predictor network embeddings, OTR assigns rewards by computing Wasserstein distances between trajectories. This hybrid approach replaces ReLOAD's MSE-based rewards with OT-based distances, hypothesizing that OT's ability to capture geometric relationships between state distributions might better distinguish expert-like transitions. Specifically, we compute the Sinkhorn divergence (Cuturi, 2013) between the target-predictor embeddings of agent transitions and expert demonstrations, using this as the intrinsic reward signal:

Theoretically, OT-based rewards provide a more holistic measure of similarity by considering the global structure of the embedding space, whereas MSE focuses on pointwise discrepancies. However, OT introduces computational overhead due to iterative Sinkhorn iterations, scaling quadratically with dataset size. Additionally, OT's sensitivity to the choice of ground cost (e.g., Euclidean vs. cosine) necessitates careful tuning, which may limit its plug-and-play adaptability compared to ReLOAD's parameter-free MSE.

As shown in Table 7, ReLOAD+OTR achieves competitive but inconsistent performance on AntMaze tasks. While it outperforms standalone OTR in most cases, it generally underperforms pure ReLOAD. For instance, on antmaze-large-play, ReLOAD+OTR scores 58.0 ± 11.9 versus ReLOAD's 61.6 ± 12.0, highlighting a trade-off between distributional alignment and computational efficiency. The marginal gains in antmaze-medium-diverse ( 76.3 ± 14.1 → 75.0 ± 18.6) suggest that OT's benefits are task-dependent and may not justify its costs in simple environments.

The ReLOAD+OTR hybrid demonstrates that distributional alignment can marginally improve performance in specific regimes but fails to consistently outperform the original framework. This reaffirms ReLOAD's core advantage: its ability to distill effective rewards without complex optimization.

# E    Effect of Predictor Network Training Quality

To evaluate the importance of predictor training quality in ReLOAD, we conduct an ablation study comparing three regimes: (1) an under-trained predictor, trained for only a few epochs on expert transitions; (2) the standard ReLOAD setup, where the predictor is trained to convergence; and (3) an over-trained predictor, further optimized beyond convergence. Each configuration is used to compute intrinsic rewards for offline RL training using IQL.

As shown in Table 8, performance degrades when using an under-trained predictor across all tasks, with a more significant drop on the replay and expert datasets. This degradation occurs because insufficiently trained predictors fail to produce meaningful prediction errors, thereby introducing noise into policy learning. The impact is particularly highlighted when the offline dataset contains expert trajectories. In such cases, a poorly trained predictor cannot effectively distinguish between expert and non-expert transitions, undermining the ability of the reward model to guide learning. This highlights the importance of sufficiently training the predictor network to ensure that prediction errors reflect similarity to expert behavior.

Table 8: Ablation on predictor network training quality. ReLOAD uses a predictor trained to convergence on expert transitions. Undertraining leads to degraded performance, especially in medium-replay settings. Overtraining yields similar results to ReLOAD, confirming robustness.

| Dataset | Undertrained | ReLOAD (Standard) | Overtrained |
|---|---|---|---|
| halfcheetah-medium | $40.4 \pm 1.3$ | $48.2 \pm 0.7$ | $45.7 \pm 0.6$ |
| halfcheetah-medium-replay | $34.3 \pm 2.6$ | $42.6 \pm 0.8$ | $42.9 \pm 1.1$ |
| halfcheetah-medium-expert | $40.8 \pm 1.4$ | $92.6 \pm 2.5$ | $92.3 \pm 2.2$ |
| walker2d-medium | $68.2 \pm 0.7$ | $81.3 \pm 0.9$ | $81.2 \pm 1.9$ |
| walker2d-medium-replay | $55.9 \pm 3.9$ | $78.6 \pm 2.5$ | $76.2 \pm 2.6$ |
| walker2d-medium-expert | $67.6 \pm 1.4$ | $110.5 \pm 0.7$ | $110.9 \pm 0.9$ |

Table 9: Ablation study on the effect of removing the squashing function from ReLOAD. Removing the squashing function leads to performance degradation, especially in replay datasets.

| Dataset | ReLOAD (Standard) | ReLOAD (No Squash) |
|---|---|---|
| halfcheetah-medium | $48.2 \pm 0.7$ | $45.09 \pm 0.6$ |
| halfcheetah-medium-replay | $42.6 \pm 0.8$ | $42.35 \pm 0.6$ |
| halfcheetah-medium-expert | $92.6 \pm 2.5$ | $88.00 \pm 3.0$ |
| walker2d-medium | $81.3 \pm 0.9$ | $41.40 \pm 9.2$ |
| walker2d-medium-replay | $78.6 \pm 2.5$ | $0.70 \pm 1.0$ |
| walker2d-medium-expert | $110.5 \pm 0.7$ | $94.52 \pm 27.6$ |

In contrast, both the standard and over-trained predictors achieve consistently high performance, suggesting that ReLOAD is robust to mild overfitting of the predictor. Once the predictor has minimized its prediction error on expert data, the reward model provides stable and informative signals for policy learning.

## F  Effect of the Squashing Function

As shown in Table 9, removing the squashing function leads to inconsistent and often degraded performance, particularly on the replay datasets. For instance, in walker2d-medium-replay, performance drops sharply to $0.70 \pm 1.0$, while in walker2d-medium it falls from $81.3 \pm 0.9$ to $41.40 \pm 9.2$. Even in expert settings like walker2d-medium-expert, where the score remains relatively high, the variance increases substantially.

As detailed in Section 4, ReLOAD transforms the raw prediction error using an exponential squashing function applied to its negative. This ensures that transitions predicted well by the expert-trained model receive higher intrinsic rewards, while poorly predicted transitions are smoothly down-weighted. In contrast, using the negative error directly (i.e., without squashing) results in unbounded and highly variable reward scales, which can destabilize training—especially in mixed-quality datasets like medium-replay.

These findings highlight that the squashing function is not merely a numerical convenience but a crucial regularizer in ReLOAD. It stabilizes the reward signal and improves robustness by preventing large prediction errors from dominating the intrinsic reward landscape.

# G    Effect of $\alpha$ AND $\beta$

Table 10: Effect of reward squashing parameters on the performance of ReLOAD.

| Environment | ReLOAD ($\alpha = 5, \beta = 0.5$) | ReLOAD ($\alpha = 10, \beta = 5$) |
|---|---|---|
| halfcheetah-medium | $46.5 \pm 0.8$ | $48.2 \pm 0.65$ |
| halfcheetah-medium-replay | $43 \pm 1.6$ | $42.6 \pm 0.8$ |
| halfcheetah-medium-expert | $92 \pm 2.8$ | $92.6 \pm 2.52$ |
| walker2d-medium | $77.3 \pm 5.0$ | $81.3 \pm 0.96$ |
| walker2d-medium-replay | $76.8 \pm 6.5$ | $78.6 \pm 2.5$ |
| walker2d-medium-expert | $108.1 \pm 3.8$ | $110.46 \pm 0.7$ |
| pen-cloned | $64.5 \pm 24.6$ | $64.5 \pm 24.6$ |
| pen-human | $88.3 \pm 8.7$ | $88.3 \pm 8.7$ |

# H    Additional Experiment

This section compares ReLOAD with Behavior Cloning (BC), a traditional imitation learning method that directly learns a policy from expert demonstrations without incorporating environmental dynamics or rewards. Table 11 highlights ReLOAD's superior performance over BC in most environments within the MuJoCo Locomotion Dataset.

Table 11: Performance Comparison of BC-10 with ReLOAD on MuJoCo Locomotion Dataset

| Environment | BC | ReLOAD |
|---|---|---|
| halfcheetah-medium | 42.5 | $48.2 \pm 0.65$ |
| halfcheetah-medium-replay | 40.6 | $42.6 \pm 0.8$ |
| halfcheetah-medium-expert | 92.9 | $92.6 \pm 2.52$ |
| hopper-medium | 56.9 | $78.72 \pm 6.9$ |
| hopper-medium-replay | 75.9 | $95.86 \pm 2.2$ |
| hopper-medium-expert | 110.9 | $104.8 \pm 9.0$ |
| walker2d-medium | 75.0 | $81.3 \pm 0.96$ |
| walker2d-medium-replay | 62.5 | $78.6 \pm 2.5$ |
| walker2d-medium-expert | 109.0 | $110.46 \pm 0.7$ |

Table 12: Performance Comparison of IQL and ReLOAD on D4RL Locomotion Tasks with varying amounts of offline data. Reported scores are normalized returns (mean $\pm$ std) over 10 seeds.

| DatasetDatasetDatasetpt< -Datasetpt> | Data fraction 10% | | Data fraction 25% | | Data fraction 50% | | Data fraction 100% | |
|---|---|---|---|---|---|---|---|---|
| | IQL | ReLOAD | IQL | ReLOAD | IQL | ReLOAD | IQL | ReLOAD |
| halfcheetah-medium | 45.6±0.9 | 45.8±0.8 | 47.2±0.2 | 47.4±0.4 | 47.4±0.2 | 47.7±0.4 | 47.4±0.2 | 48.2±0.65 |
| hopper-medium | 53.4±5.0 | 63.8±7.8 | 55.8±7.5 | 63.68±10.7 | 51.4±5.4 | 65.7±4.3 | 66.2±5.7 | 78.72±6.9 |
| walker2d-medium | 62.4±14.2 | 78.2±5.9 | 62.2±7.4 | 78.2±5.8 | 63.9±7.4 | 79.8±1.8 | 78.3±8.7 | 81.3±0.96 |

Table 13: Performance Comparison on D4RL Locomotion Tasks. The reported results are normalized scores (mean ± standard deviation) across over 10 random seeds. The results correspond to K = 5 expert trajectories. The highest scores are highlighted in green.

| Dataset | IQL | OTR (K=5) | CLUE (K=5) | AILOT (K=5) | ReLOAD (K=5) |
|---|---|---|---|---|---|
| halfcheetah-medium | $47.4 \pm 0.2$ | $43.3 \pm 0.2$ | $45.2 \pm 0.2$ | $46.6 \pm 0.2$ | $47.2 \pm 0.9$ |
| halfcheetah-medium-replay | $44.2 \pm 1.2$ | $41.9 \pm 0.3$ | $43.2 \pm 0.4$ | $41.2 \pm 0.5$ | $43.5 \pm 0.9$ |
| halfcheetah-medium-expert | $86.7 \pm 5.3$ | $89.9 \pm 1.9$ | $91.4 \pm 1.4$ | $92.4 \pm 1.0$ | $92.8 \pm 1.7$ |
| hopper-medium | $66.2 \pm 5.7$ | $79.5 \pm 5.3$ | $79.1 \pm 3.5$ | $82.5 \pm 3.7$ | $78.72 \pm 6.9$ |
| hopper-medium-replay | $94.7 \pm 8.6$ | $85.4 \pm 1.7$ | $93.3 \pm 4.5$ | $97.4 \pm 0.1$ | $95.8 \pm 2.3$ |
| hopper-medium-expert | $91.5 \pm 14.3$ | $90.4 \pm 21.5$ | $104.0 \pm 5.4$ | $107.3 \pm 5.6$ | $110.4 \pm 5.0$ |
| walker2d-medium | $78.3 \pm 8.7$ | $79.8 \pm 1.4$ | $79.6 \pm 0.7$ | $80.9 \pm 1.4$ | $81.0 \pm 2.5$ |
| walker2d-medium-replay | $73.8 \pm 7.1$ | $71.0 \pm 5.0$ | $75.1 \pm 1.3$ | $76.9 \pm 1.6$ | $77.2 \pm 2.5$ |
| walker2d-medium-expert | $109.6 \pm 1.0$ | $109.4 \pm 0.4$ | $109.9 \pm 0.4$ | $110.3 \pm 0.4$ | $110.4 \pm 0.7$ |
| D4RL Locomotion total | 692.4 | 690.6 | 721.3 | 735.5 | 737.2 |

