# OpenReview forum: "From Novelty to Imitation: Self-Distilled Rewards for Offline Reinforcement Learning"
_TMLR — Accepted by TMLR_

### Review · Reviewer_GS7r · 2025-06-20

**Summary Of Contributions:**

This paper presents a novel reward annotation framework for offline RL. The method is based on Random Network Distillation which is simple and straightforward. The experiment results on multiple RL tasks demonstrate the successfulness of the proposed method.

**Audience:**

Yes

**Broader Impact Concerns:**

I don't see obvious impact concerns from this submission.

**Claims And Evidence:**

Yes

**Requested Changes:**

I am confused by some small notation issues, and would love authors to check it and maybe it is my misunderstanding.
- what is D_a? is it different from D_U?
- In Sec 5 around Eq. 17 and Eq. 18, I am wondering why use f(s_i, s_i') in Eq. 17 but use x_j=(s_j, s_j') in Eq. 18? I think these uses of notations can be synced.
- In Sec. 6, it is unclear what is the "state-of-the-art approaches" here means? Do you mean online or offline RL methods?

**Strengths And Weaknesses:**

Strength:
- The method is simple and effective according to the experiment result.
- This paper provide discussion and justification with the experiment results, which is thorough for explaining the effectiveness of the proposed method.

Weakness:
- It is a bit unclear what is the size of the training data of the predictor network? In the experiment section, I think it only said from 1-5 expert demonstrations, but how many transitions are included? I am a bit curious whether a small number of samples can really train the model effectively. And the embedding space might not be meaningful enough.
- I have to admit I am not an expert of RL, and that is why I somehow feel that the technical novel is somehow thin. The proposed method is reusing previously proposed concepts (RND) directly. I am not saying for rejecting this paper, but I sort of want to know more on how impactful this method will be by checking authors response and other reviewers comments.

---

> ### Author Response · Authors · 2025-07-31
> **Response to GS7r**
>
> We thank the reviewer for their thoughtful comments and appreciate the opportunity to clarify both technical aspects and notational points. We address each issue below:
>
> **Weakness 1:**
> As you correctly noted, the predictor is trained on a small number of expert demonstrations—specifically between 1 to 5 trajectories, depending on the experiment. The number of transitions in each trajectory depends on the task. For example, a half-cheetah trajectory typically consists of 1000 transitions.
>
> Despite the limited data, we observe that the predictor effectively learns to represent expert behavior, as evidenced by the strong downstream performance of the learned reward. To further validate this, we have added an ablation study in Appendix E, analyzing the effect of training the predictor under different levels of convergence (undertrained vs. standard vs. overtrained). The results confirm that even with limited expert data, the predictor can guide offline RL effectively, provided it captures the expert structure.
> | **Dataset**                | **Undertrained**    | **ReLOAD (Standard)** | **Overtrained**       |
> |---------------------------|---------------------|-----------------------|------------------------|
> | walker2d-medium           | $68.2 \pm 0.7$      | $81.3 \pm 0.9$        | $81.2 \pm 1.9$         |
> | walker2d-medium-replay    | $55.9 \pm 3.9$      | $78.6 \pm 2.5$        | $76.2 \pm 2.6$         |
> | walker2d-medium-expert    | $67.6 \pm 1.4$      | $110.5 \pm 0.7$       | $110.9 \pm 0.9$        |
>
> **weakness 2:** We have now explicitly clarified the conceptual and technical differences between ReLOAD and [1]. These distinctions are detailed in our common response to all reviewers, and also incorporated directly into the main paper (see Section 1, the clarified framing in Section 4, and Section 7).
> In short, while both methods use RND, ReLOAD leverages prediction error over state transitions as a proxy for expert-likeness, whereas prior work uses it as a novelty penalty over state-action pairs. This leads to a fundamentally different use case, training setup, and theoretical motivation, which we believe marks a meaningful contribution to the literature.
>
> **Requested Changes**
>
> **Difference between $\mathcal{D}_a$ and $D_U$:** We appreciate the reviewer’s attention to clarity. In Section 3.1, we define $\mathcal{D}_a$ as the offline dataset (a finite sample), and $D_U$ as the underlying distribution it is drawn from.
>
> **Inconsistent notation:** We thank the reviewer for pointing out the potential confusion around the notation in Equations 17 and 18. The use of different subscripts—$i$ in Eq. 17 and $j$ in Eq. 18—is intentional and reflects that the predictor network $f$ is trained on expert transitions $(s_i, s_i') \in \mathcal{D}_E$ and then applied to unlabeled offline transitions $(s_j, s_j') \in \mathcal{D}_a$. We have clarified this in Section 5 by explicitly stating that the change in indexing denotes a shift in the data distribution—from expert demonstrations to broader offline data. This distinction has now been made explicit in the revised text to improve clarity.
>
> **State-of-the-art approaches" in Section 6**
> We agree that this could be ambiguous. In the revised text, we now specify that we compare with offline RL approaches.

---

### Review · Reviewer_mEAa · 2025-07-01

**Summary Of Contributions:**

RELOAD is a simple but effective reward annotation framework for offline RL. It utilizes the principles of Random Network Distillation (RND) to generate intrinsic rewards, aiding datasets that originally lack reward signals. The results show that RELOAD achieves an improvement over previous offline RL methods on the D4RL benchmark.

**Audience:**

Yes

**Broader Impact Concerns:**

No concern.

**Claims And Evidence:**

No

**Requested Changes:**

I would appreciate a detailed, evidence-backed response to the weaknesses raised.

**Strengths And Weaknesses:**

### Strengths:

- The paper is well-written and easy to read.

- RELOAD is straightforward and effective.

- The experiments are extensive and well-conducted.

### Weaknesses:
- I would like to ask about the distinction between the approach of generating intrinsic rewards using RND followed by offline RL learning and previous methods where RND intrinsic rewards were directly integrated into offline RL training, such as in [1]. What are the fundamental differences between these two approaches? What is the unique advantage of RELOAD, and is there any supporting evidence for this claim?

- While RELOAD is presented as a plug-and-play method, I believe there are some issues with the experimental settings. It would be more convincing to compare RELOAD against existing baselines (e.g., IQL, CQL, DT) first, and then incorporate RELOAD into these baselines to demonstrate its advantages, rather than using IQL + RELOAD as a comparison against other original RL baselines. This comparison seems unfair.

- A key issue with RND is that if the predictor network fails to converge, the generated intrinsic reward may reflect this non-convergence rather than an indication of an intrinsically motivating reward. Therefore, I am particularly interested in seeing a comparison of the effects of rewards generated by predictor networks under different conditions (unconverged, converged, and overfitted) when used as the dataset for offline RL training. How do these different scenarios affect the trained policies?
---
*Reference:*

[1] Anti-Exploration by Random Network Distillation. ICML 2023.

---

> ### Author Response · Authors · 2025-07-31
> **Response to Reviewer mEAa**
>
> We sincerely thank the reviewer for their thoughtful and constructive feedback. Your feedback has helped us strengthen both the positioning and empirical evaluation of ReLOAD. Below, we respond to each of your concerns in turn:
>
> **Weakness 1:** We have now explicitly clarified the conceptual and technical differences between ReLOAD and [1]. These distinctions are detailed in our common response to all reviewers, and also incorporated directly into the main paper (see Section 1, the clarified framing in Section 4, and Section 7).
> In short, while both methods use RND, ReLOAD leverages prediction error over state transitions as a proxy for expert-likeness, whereas prior work uses it as a novelty penalty over state-action pairs. This leads to a fundamentally different use case, training setup, and theoretical motivation, which we believe marks a meaningful contribution to the literature.
>
> **Weakness 2:** We appreciate your concern about ensuring fair comparisons between ReLOAD and existing offline imitation RL methods. We would like to clarify that we do include direct comparisons to vanilla baselines, such as IQL without ReLOAD, across all D4RL tasks. These results are reported in the main paper (Tables 1, 2, and 3), and they demonstrate that ReLOAD consistently outperforms standard IQL, especially in low-feedback settings (e.g., medium-replay datasets).
>
> To further support this, we also had a dedicated ablation study (Section 6.6, Figure 2, and Table 12) that compares IQL+True Rewards vs. IQL+ReLOAD side by side with varying sizes of offline data. This directly isolates the impact of the ReLOAD reward mechanism while keeping the RL algorithm fixed. The results reinforce that ReLOAD provides consistent improvements across all environments and data regimes.
>
> In addition, we emphasize that several baselines in the comparison set, including OTR, CLUE, and AILOT, also use IQL as the underlying offline RL algorithm. This makes our comparisons fair and meaningful. We have mentioned this point more explicitly in the main paper (Section 6) for better clarity.
>
> **Weakness 3:** We appreciate this insightful question. To empirically validate the importance of predictor convergence, we have added a new ablation study on predictor training quality, now included in Appendix E.
>
> This experiment evaluates performance under three conditions: undertrained, standard (converged), and overtrained predictor networks. This ablation confirms that meaningful reward generation in ReLOAD hinges on expert-aligned predictor training, and that the method is robust to moderate overtraining. A full discussion and results are provided in Appendix E of the revised submission.
>
> | **Dataset**                | **Undertrained**    | **ReLOAD (Standard)** | **Overtrained**       |
> |---------------------------|---------------------|-----------------------|------------------------|
> | walker2d-medium           | $68.2 \pm 0.7$      | $81.3 \pm 0.9$        | $81.2 \pm 1.9$         |
> | walker2d-medium-replay    | $55.9 \pm 3.9$      | $78.6 \pm 2.5$        | $76.2 \pm 2.6$         |
> | walker2d-medium-expert    | $67.6 \pm 1.4$      | $110.5 \pm 0.7$       | $110.9 \pm 0.9$        |
>
> We hope these clarifications address your concerns. The updated manuscript clearly articulates how ReLOAD differs from prior RND-based methods, clarifies the fairness of baseline comparisons, and includes new ablations validating key design choices. We sincerely thank you for your thoughtful feedback.
>
> [1] Anti-Exploration by Random Network Distillation. ICML 2023.

---

> > ### Comment · Reviewer_mEAa · 2025-08-29
> > **Response to the rebuttal**
> >
> > Thank you for your reply. I think the results are interesting. And I am satisfied with the results.

---

### Review · Reviewer_Ro2J · 2025-07-24

**Summary Of Contributions:**

The authors investigate the problem of reward annotation in reward-free offline reinforcement learning.

**1. Reward function**
The reward function is introduced by computing the prediction error between a predictor network and a target network on expert demonstrations.

**2. Justification of RND rewards**
Introducing a rigorous theory for the effectiveness of the RND reward. (I didn't check the proofs in details)

**3. Indirect transition similarity measurement**
A high reward indicates the lower similarity between expert transitions and transitions from the offline dataset.

**4. Reward Squashing**
Providing a reward squashing technique bounding reward variance for improving training stability.

**5. Extensive experimental evaluations on multiple benchmarks**
- Performing multiple experiments on different offline RL benchmarks, including D4RL Locomotion, Antmaze, and Adroit.
- Discussing the reduced computation time compared to AILOT.
- ReLoad achieves better performance than baselines on D4RL and Adroit tasks, while being competitive on Antmaze tasks.
- ReLoad performs better across 2 different expert dataset sizes and 4 different offline dataset sizes than baselines.

**Audience:**

Yes

**Claims And Evidence:**

Yes

**Requested Changes:**

1. Clarifying the connections and differences to the related works, RND and the work (Nikulin et al., 2023).
1. Providing implementation details of the target and the predictor networks, including network architecture and parameter initialization.
2. Providing ablations on the reward squashing technique, e.g., comparing ReLoaD to ReLoaD without reward squashing.

**Strengths And Weaknesses:**

**Strengths**
1. The reward function is simple, motivated, and easily computational.
2. The experiments are conducted over multiple datasets.
3. The proposed method achieves strong performance.
4. Providing ablation studies on the expert and offline dataset sizes.

**Weakness**
1. The main problem for me is that the reward function, which is the main contribution of this work, was proposed by RND and already applied in offline RL (Nikulin et al., 2023). This greatly weakens this work in terms of theoretical and technical contributions.
2. The authors don't discuss the differences to the related works, RND and the work (Nikulin et al., 2023). Related discussions on the differences can better position this work and thus are highly encouraged.
2. I don't understand how the prediction error works. The prediction error is the difference between outputs of the predictor network and the target network taking the same inputs. If these two networks have the same network architecture and initialized parameters, the prediction error will be always zero in the beginning of training. As a result, the prediction error cannot be used for transition assessment.
3. It is claimed that the reward squashing technique is effective in improving training stability. The corresponding empirical results are missed for supporting the claim.

---

> ### Author Response · Authors · 2025-07-31
> **Response to Reviewer Ro2J**
>
> We sincerely thank the reviewer for their thoughtful and detailed feedback. Your comments helped us significantly improve the clarity and positioning of our contributions. Below, we address each concern point by point:
>
> **Requested changes 1 and weakness 1\&2:** We have now explicitly clarified the conceptual and technical differences between ReLOAD and [1]. These distinctions are detailed in our common response to all reviewers, and also incorporated directly into the main paper (see Section 1, the clarified framing in Section 4, and Section 7).
> In short, while both methods use RND, ReLOAD leverages prediction error over state transitions as a proxy for expert-likeness, whereas prior work uses it as a novelty penalty over state-action pairs. This leads to a fundamentally different use case, training setup, and theoretical motivation, which we believe marks a meaningful contribution to the literature.
>
> **Requested changes 2 and weakness 3:** We thank the reviewer for this insightful question. We initialize both the predictor and target networks in RND separately. Although the predictor and target networks share the same architecture, they are initialized independently, resulting in a random prediction error at the start. This error becomes meaningful as the predictor is trained solely on expert transitions, learning to minimize the error only on expert-like patterns.
>
> As a result, the prediction error evolves to reflect how well the predictor has learned to approximate the target network's outputs on the expert data. Transitions that resemble expert ones yield low prediction error, while unfamiliar or out-of-distribution transitions result in higher error, since the predictor has not been trained on those regions of the state space. This gap in prediction is precisely what we exploit as a reward signal.
>
> To clarify this empirically, we have added an ablation study (see Appendix E, Table 8) that analyzes the impact of predictor training quality on downstream RL performance. The results show that the learned reward is only effective when the predictor has been sufficiently trained on expert transitions. This confirms that prediction error becomes a meaningful discriminator between expert-like and non-expert transitions once training progresses.
>
> | **Dataset**                | **Undertrained**    | **ReLOAD (Standard)** | **Overtrained**       |
> |---------------------------|---------------------|-----------------------|------------------------|
> | walker2d-medium           | $68.2 \pm 0.7$      | $81.3 \pm 0.9$        | $81.2 \pm 1.9$         |
> | walker2d-medium-replay    | $55.9 \pm 3.9$      | $78.6 \pm 2.5$        | $76.2 \pm 2.6$         |
> | walker2d-medium-expert    | $67.6 \pm 1.4$      | $110.5 \pm 0.7$       | $110.9 \pm 0.9$        |
>
> Additionally, in response to your suggestion, we have clarified the implementation details of both the predictor and target networks in Appendix Section A, Table 4, and revised the language to improve readability and precision.
>
> **Requested changes 3 and weakness 4:**
> We thank the reviewer for pointing out the missing empirical support for the reward squashing technique. To address this, we have conducted and included an additional ablation study in Section F of the Appendix, which compares ReLOAD with and without the reward squashing step. The results (see Appendix F, Table 9) demonstrate that removing the squashing mechanism leads to severe reward explosion and instability, especially on replay datasets:
>
> | **Dataset**                | **ReLOAD (Standard)** | **ReLOAD (No Squash)**   |
> |---------------------------|------------------------|---------------------------|
> | walker2d-medium           | $81.3 \pm 0.9$         | $41.40 \pm 9.2$           |
> | walker2d-medium-replay    | $78.6 \pm 2.5$         | $0.70 \pm 1.0$            |
> | walker2d-medium-expert    | $110.5 \pm 0.7$        | $94.52 \pm 27.6$          |
>
>
> These results validate the importance of reward squashing in stabilizing training across datasets. A more detailed discussion is now included in Appendix F for completeness.
>
> We hope these additions and clarifications comprehensively address your concerns. The updated manuscript now clearly explains how ReLOAD differs from prior RND work, provides implementation transparency, and includes empirical studies to justify core design choices. We thank you again for helping us improve the clarity and rigor of our submission.

---

> > ### Author Response · Authors · 2025-07-31
> >
> > [1] Anti-Exploration by Random Network Distillation. ICML 2023.

---

> > > ### Comment · Reviewer_Ro2J · 2025-08-11
> > > **Thank you**
> > >
> > > Thank the authors for addressing my concerns. I have no further questions.

---

### Author Response · Authors · 2025-07-31
**Response to all reviewers: Distinction Between ReLOAD and Anti Exploration by RND**

We appreciate reviewers' questions regarding the novelty of our work relative to prior RND-based methods, particularly, "Anti-Exploration by Random Network Distillation [1]." Below, we provide a point-wise clarification outlining the conceptual and technical distinctions between ReLOAD and [1]:

**Objective and Paradigm Difference:** [1] uses Random Network Distillation (RND) in the context of conservative offline reinforcement learning, where the objective is to penalize out-of-distribution or novel state-action pairs to reduce extrapolation error. Their predictor network operates on (s, a) inputs, and high prediction error is interpreted as novelty, which is then applied as a penalty during training to discourage unfamiliar actions.

In contrast, ReLOAD operates in a completely different paradigm: reward-free offline imitation learning. Instead of state-action inputs, ReLOAD applies RND over state transitions (s, s'), abstracting away from actions and focusing on whether the transition dynamics resemble those observed in expert trajectories. We reinterpret RND as a reward inference mechanism by training the predictor solely on expert state transitions and treating low prediction error as a proxy for expert-likeness. Rather than discouraging novelty, ReLOAD promotes transitions aligned with expert behavior—using prediction error directly as a reward signal. This makes our approach better suited for imitation settings.

**Supervision and Training Signal:**
The predictor in [1] is trained on the full offline dataset, which may include suboptimal or noisy transitions (s, a). As such, the prediction error encodes novelty relative to the entire dataset. In ReLOAD, the predictor is trained exclusively on expert demonstrations (s, s'). This design ensures that prediction error reflects proximity to expert behavior rather than general uncertainty. Consequently, our reward signal directly favors state transitions that resemble those seen in expert trajectories.

**Reward Mechanism and Use:** In [1], the RND-derived signal is incorporated as a penalty term within the Q-function objective, effectively reducing the value of novel or unfamiliar transitions. It is not used to shape the reward function directly.
In contrast, ReLOAD uses prediction error directly as a reward function for downstream offline imitation RL. The lower the prediction error (i.e., the more expert-like the state transition), the higher the reward assigned. This fundamentally flips the role of RND: from discouraging exploration to enabling imitation.

**Theoretical Foundations**
[1] Do not provide a formal theoretical analysis linking RND error to expert support or transition quality. ReLOAD fills this gap with Theorem 1, which establishes that under standard assumptions, expert state transitions exhibit strictly lower prediction error than non-expert ones. This provides a rigorous justification for using prediction error as a reward proxy for imitation learning, something not explored in prior RND-based approaches.

**Empirical Setting and Generality:** The motivation for [1] is to improve robustness in fully offline RL with known rewards by avoiding risky, out-of-distribution actions. ReLOAD, by contrast, addresses a more challenging and less explored setting: reward-free offline imitation, where no external reward labels are available and assumes no access to expert actions. Additionally, while [1] is tightly integrated into a particular loss function, ReLOAD is designed as a plug-and-play module that can be combined with any offline RL algorithm. It requires only a small set of expert demonstrations and scales across domains.

**Summary:** In summary, while both approaches leverage RND, they differ fundamentally in their goals, supervision strategies, input representations, and theoretical motivations. ReLOAD introduces a novel use of RND as a reward inference mechanism tailored for offline imitation learning under minimal supervision—without requiring access to extrinsic rewards or expert actions. By aligning prediction error with expert-like transition dynamics, ReLOAD enables scalable and robust offline imitation learning in challenging reward-free settings. We have made these distinctions explicit in the updated manuscript (Section 1: Introduction, Section 4: Method, and Section 7: Conclusion), which further clarifies how ReLOAD departs from prior novelty-based uses of RND.


[1] Anti-Exploration by Random Network Distillation. ICML 2023.

---

### Author Response · Authors · 2025-10-07

Dear AE,

We sincerely thank the reviewers and the action editor for their thoughtful and constructive feedback, which greatly improved the clarity and quality of our manuscript. We have uploaded the camera-ready version of the manuscript.

Authors 4773

---

### Decision · Action_Editor_ztpP · 2025-10-02

**Recommendation:** Accept as is

**Audience:**

Yes

**Audience Explanation:**

All the reviewers agreed.
* Addresses a timely problem in reward-free offline RL with a simple, plug-and-play annotation mechanism.
* Practical relevance for imitation settings without reward labels or actions.

**Claims And Evidence:**

Yes

**Claims Explanation:**

All the reviewers agreed.
* Strong empirical results across D4RL locomotion, AntMaze, and Adroit.
* Added ablations on predictor training quality and reward squashing; reviewers confirmed concerns addressed.
* Clear theoretical justification linking RND error to expert-likeness.